# CAML: Fast Context Adaptation via Meta-Learning

## Abstract

We propose CAML, a meta-learning method for fast adaptation that partitions the model parameters into two parts: *context parameters* that serve as additional input to the model and are adapted on individual tasks, and *shared parameters* that are meta-trained and shared across tasks. At test time, the context parameters are updated with one or several gradient steps on a task-specific loss that is backpropagated through the shared part of the network. Compared to approaches that adjust all parameters on a new task (e.g., MAML), our method can be scaled up to larger networks without overfitting on a single task, is easier to implement, and saves memory writes during training and network communication at test time for distributed machine learning systems. We show empirically that this approach outperforms MAML, is less sensitive to the task-specific learning rate, can capture meaningful task embeddings with the context parameters, and outperforms alternative partitionings of the parameter vectors.

## 1 Introduction

A key challenge in meta-learning is fast adaptation: learning on previously unseen tasks *fast* and with *little data*. In principle, this can be achieved by leveraging knowledge obtained in other, related tasks. However, the best way to do so remains an open question.

A popular recent method for fast adaptation is *model agnostic meta learning* (MAML) (Finn et al., 2017a), which learns a *model initialisation*, such that at test time the model can be adapted to solve the new task in only a few gradient steps. MAML has an interleaved training procedure, comprised of inner loop and outer loop updates that operate on a batch of tasks at each iteration. In the inner loop, MAML learns task-specific parameters by performing one gradient step on a task-specific loss. Then, in the outer loop, the model parameters from *before* the inner loop update are updated to reduce the expected loss across tasks *after* the inner loop update on the individual tasks. Hence, MAML learns a model initialisation that, at test time, can generalise to a new task after only a few gradient updates.

However, while MAML adapts the entire model to the new task, many transfer learning algorithms adapt only a fraction of the model (Kokkinos, 2017), keeping the rest fixed across tasks. For example, representations learned for images can be transferred to different image classification domains (Donahue et al., 2014) or reused for object tracking in videos (Wojke et al., 2017). This suggests that some model parameters can be considered task independent and others task specific. Adapting only some model parameters can make learning faster and easier, as well as mitigate overfitting and catastrophic forgetting.

To this end, we propose *context adaptation for meta-learning* (CAML), a new method for fast adaptation via meta-learning. Like MAML, CAML learns a model initialisation that can quickly be adapted to new tasks. However, unlike MAML, it adapts only a subset of the model parameters to the new task. While restricting adaptation in this way is straightforward, it raises a key question: how should we decide which parameters to adapt and which to keep fixed? The main insight behind CAML is that, for many fast adaptation problems, the inner loop reduces to a task identification problem, rather than learning how to solve the whole task, which is typically infeasible with only a few gradient updates. Thus, it suffices if the part of the model that varies across tasks is an additional *input* to the model, and is independent of its other inputs.

These additional inputs, which we call *context parameters* $\phi$ (see Figure 1), can be interpreted as a task embedding that modulates the behaviour of the model. This embedding is learned via backpropagation during the inner loop of a meta-learning procedure similar to MAML, while the rest of the model is updated only in the outer loop. This allows CAML to explicitly optimise the task-independent parameters $\theta$ for good performance across tasks, while ensuring that the task-specific context parameters $\phi$ can quickly adapt to new tasks.

This separation of task solver and task embedding has several advantages. First, the size of both components can be chosen appropriately for the task. In particular, the network can be made expressive enough without overfitting to a single task in the inner loop, which we show empirically MAML is prone to. Model design and architecture choice also benefit from this separation, since for many practical problems we have prior knowledge of which aspects vary across tasks and hence how much capacity the context parameter $\phi$ should have. Like MAML, our method is model-agnostic, i.e., it can be applied to any model that is trained via gradient descent. However, CAML is easier to implement: assigning the correct computational graphs for higher order gradients is done only on the level of the context parameters, avoiding manual access and operations on the network weights and biases. Furthermore, parameter copies are not necessary which saves memory writes, a common bottleneck for running on GPUs. CAML can also help distributed machine learning systems, where the same model is deployed to different machines and we wish to learn different contexts concurrently. Network communication is often the bottleneck, which is mitigated by only sharing the (gradients of) context parameters.

We show empirically that CAML outperforms MAML on a regression and classification task and performs similarly on a reinforcement learning problem, while adapting significantly fewer parameters at test time. We observe that CAML is less sensitive to the inner-loop learning rate, and can be scaled up to larger networks without overfitting. We also demonstrate that the context parameters represent meaningful embeddings of tasks, confirming that the inner loop acts as a task identification step.

## 2 BACKGROUND: META-LEARNING FOR FAST ADAPTATION

We consider settings where the goal is to learn models that can quickly adapt to a new task with only little data. To this end, learning on the new task is preceded by meta-learning on a set of related tasks. Here we describe the meta-learning problem for supervised and reinforcement learning, as well as the method MAML.

### 2.1 PROBLEM SETTING

In few-shot learning problems, we are given distributions over training tasks $p_{\text{train}}(\mathcal{T})$ and test tasks $p_{\text{test}}(\mathcal{T})$. Training tasks can be used to learn how to adapt fast to any of the tasks with little per-task data, and evaluation is done on previously unseen test tasks. Unless stated otherwise, we assume that $p_{\text{train}} = p_{\text{test}}$ and refer to both as $p$. Tasks in $p$ typically share some structure, so that transferring knowledge between tasks speeds learning. During each meta-training iteration, a batch of $N$ tasks $\mathbf{T} = \{\mathcal{T}_i\}_{i=1}^{N}$ is sampled from $p$.

**Supervised Learning.** In a supervised learning setting, we learn a model $f$ that maps data points $x \in \mathcal{X}$ that have a true label $y \in \mathcal{Y}$ to predictions $\hat{y} \in \mathcal{Y}$. A task $\mathcal{T}_i$ is defined as a tuple $\mathcal{T}_i = (\mathcal{X}, \mathcal{Y}, \mathcal{L}, q)$, where $\mathcal{X}$ is the input space, $\mathcal{Y}$ is the output space, $\mathcal{L}(y, \hat{y})$ is a task-specific loss function, and $q(x, y)$ is a distribution over labelled data points. We assume that all data points are drawn i.i.d. from $q$. Different tasks can be created by changing any element of $\mathcal{T}_i$.

Training in the supervised meta-learning setting proceeds over meta-training iterations, where for every task $\mathcal{T}_i \in \mathbf{T}$ from the current batch, we sample two datasets $\mathcal{D}_i^{\text{train}}$ (for training) and $\mathcal{D}_i^{\text{test}}$ from $q_{\mathcal{T}_i}$:

$$\mathcal{D}_i^{\text{train}} = \{(x,y)^{i,m}\}_{m=1}^{M_i^{\text{train}}}, \qquad \mathcal{D}_i^{\text{test}} = \{(x,y)^{i,m}\}_{m=1}^{M_i^{\text{test}}}, \tag{1}$$

where $(x, y) \sim q_{\mathcal{T}_i}$. $M_i^{\text{train}}$ and $M_i^{\text{test}}$ are the number of training and test datapoints, respectively. The training data is used to update $f$, and the test data is then used to evaluate how good this update was, and adjust $f$ or the update rule accordingly.

**Reinforcement Learning.** In a reinforcement learning (RL) setting, we aim to learn a policy $\pi$ that maps states $s \in \mathcal{S}$ to actions $a \in \mathcal{A}$. Each task corresponds to a *Markov decision process* (MDP): a tuple $\mathcal{T}_i = (\mathcal{S}, \mathcal{A}, r, q, q_0)$, where $\mathcal{S}$ is a set of states, $\mathcal{A}$ is a set of actions, $r(s_t, a_t, s_{t+1})$ is a reward function, $q(s_{t+1}|s_t, a_t)$ is a transition function, and $q_0(s_0)$ is an initial state distribution. The goal is to maximise the expected cumulative reward $\mathcal{J}$ under $\pi$,

$$\mathcal{J}(\pi) = \mathbb{E}_{q_0, q, \pi} \left[ \sum_{t=0}^{H-1} \gamma^t r(s_t, a_t, s_{t+1}) \right], \tag{2}$$

where $H \in \mathbb{N} \cup \infty$ is the horizon and $\gamma \in [0, 1]$ is the discount factor. Again, different tasks can be created by changing any element of $\mathcal{T}_i$.

During each meta-training iteration, for every task $\mathcal{T}_i \in \mathbf{T}$ from the current batch, we first collect a trajectory

$$\tau_i^{\text{train}} = \left\{ s_0, a_0, r_0, s_1, a_1, r_1, \ldots, s_{M_i^{\text{train}}-1}, a_{M_i^{\text{train}}-1}, r_{M_i^{\text{train}}-1}, s_{M_i^{\text{train}}} \right\}, \tag{3}$$

where the initial state $s_0$ is sampled from $q_0$, the actions are chosen by the current policy $\pi$, the state transitions according to $q$, and $M_i^{\text{train}}$ is the number of environment interactions available, We unify several episodes in this formulation: if the horizon $H$ is reached within the trajectory, the environment is reset using $q_0$. Once the trajectory is collected, this data is used to update the policy. Another trajectory $\tau_i^{\text{test}}$ is then collected by rolling out the updated policy for $M_i^{\text{test}}$ time steps. This test trajectory is used to evaluate the quality of the update on that task, and to adjust $\pi$ or the update rule accordingly.

Evaluation for both supervised and reinforcement learning problems is done on a new (unseen) set of tasks drawn from $p$ (or $p_{\text{test}}$ if the test distribution of task is different). For each such task, the model is updated using $\mathcal{L}$ or $\mathcal{J}$ and only few datapoints ($\mathcal{D}^{\text{train}}$ or $\tau^{\text{train}}$). Performance of the updated model is reported on $\mathcal{D}^{\text{test}}$ or $\tau^{\text{test}}$.

## 2.2 Model-Agnostic Meta-Learning

One method for few-shot learning is *model-agnostic meta-learning* (Finn et al., 2017a, MAML). Here, we describe the application of MAML to a supervised learning setting. MAML learns an initialisation for the parameters $\theta$ of a model $f_\theta$ such that, given a new task, a good model for that task can be learned with only a small number of gradient steps and data points. In the inner loop, MAML computes new task-specific parameters $\theta_i$ (starting from $\theta$) via one[1] gradient update,

$$\theta_i = \theta - \alpha \nabla_\theta \frac{1}{M_{\text{train}}^i} \sum_{(x,y) \in \mathcal{D}_i^{\text{train}}} \mathcal{L}_{\mathcal{T}_i}(f_\theta(x), y). \tag{4}$$

For the meta-update in the outer loop, the *original* model parameters are then updated with respect to the performance after the inner-loop update, i.e.,

$$\theta \leftarrow \theta - \beta \nabla_\theta \frac{1}{N} \sum_{\mathcal{T}_i \in \mathbf{T}} \frac{1}{M_{\text{test}}^i} \sum_{(x,y) \in \mathcal{D}_i^{\text{test}}} \mathcal{L}_{\mathcal{T}_i}(f_{\theta_i}(x), y). \tag{5}$$

The result of training is a model initialisation $\theta$ that can be adapted with just a few gradient steps to any new task that we draw from $p$. Since the gradient is taken with respect to the parameters $\theta$ before the inner-loop update (4), the outer-loop update (5) involves higher order derivatives in $\theta$.

## 3 Fast Context Adaptation via Meta-Learning

We propose to partition the model parameters into two parts: context parameters $\phi$ that are adapted in the inner loop on an individual task, and parameters $\theta$ that are shared across tasks and meta-learned in the outer loop. In the following we describe the training procedure for supervised and reinforcement learning problems. Pseudo-code is provided in Appendix A.

### 3.1 Supervised Learning

At every meta-training iteration and for the current batch $\mathbf{T}$ of tasks, we use the training data $\mathcal{D}_i^{\text{train}}$ as follows. Starting from $\phi_0$, which can either be fixed or meta-learned as well (we often choose $\phi_0 = 0$; see Section 3.4), we learn task-specific parameters $\phi_i$ via one gradient update:

$$\phi_i = \phi_0 - \alpha \nabla_\phi \frac{1}{M_i^{\text{train}}} \sum_{(x,y) \in \mathcal{D}_i^{\text{train}}} \mathcal{L}_{\mathcal{T}_i}(f_{\phi_0, \theta}(x), y). \tag{6}$$

While we only take the gradient with respect to $\phi$, the updated parameter $\phi_i$ is also a function of $\theta$, since during backpropagation, the gradients flow through the model. Once we have collected the updated parameters $\phi_i$ for all sampled tasks, we proceed to the meta-learning step, in which $\theta$ is updated:

$$\theta \leftarrow \theta - \beta \nabla_\theta \frac{1}{N} \sum_{\mathcal{T}_i \in \mathbf{T}} \frac{1}{M_i^{\text{test}}} \sum_{(x,y) \in \mathcal{D}_i^{\text{test}}} \mathcal{L}_{\mathcal{T}_i}(f_{\phi_i, \theta}(x), y). \tag{7}$$

This update includes higher order gradients in $\theta$ due to the dependency on (6).

---

[1] We outline the method for one gradient update here, but several gradient steps can be performed at this point as well.

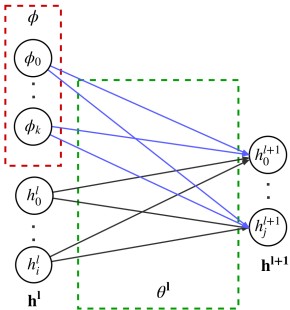

Figure 1: **Context adaptation.** A network layer $h^l$ is augmented with additional context parameters $\phi$ (red), which are initialised to 0 before each adaptation step. The context parameters are updated by gradient descent during each inner loop and during test time. The network parameters $\theta$ (green) are only updated in the outer loop and shared across tasks. Hence, they stay fixed at test time. By initialising $\phi$ to 0, the network parameters associated with the context parameters (blue) do not affect the output of the layer before adaptation. After the first adaptation step they are used to modulate the rest of the network in order to solve the new task.

### 3.2 Reinforcement Learning

During each iteration, for a current batch of MDPs $\mathbf{T} = \{\mathcal{T}_i\}_{i=1}^N$, we proceed as follows. Given $\phi_0$ (see Section 3.4), we collect a rollout $\tau_i^{\text{train}}$ by executing the policy $\pi_{\phi_0,\theta}$. We then compute task-specific parameters $\phi_i$ via one gradient update:

$$\phi_i = \phi_0 + \alpha \nabla_\phi \tilde{\mathcal{J}}_{\mathcal{T}_i}(\tau_i^{\text{train}}, \pi_{\phi_0,\theta}), \tag{8}$$

where $\tilde{\mathcal{J}}(\tau, \pi)$ is an objective function given by any gradient-based reinforcement learning method that uses trajectories $\tau$ produced by a parameterised policy $\pi$ to update that policy's parameters, such as TRPO (Schulman et al., 2015) or DQN (Mnih et al., 2015). After updating the policy, we collect another trajectory $\tau_i^{\text{test}}$ to evaluate the updated policy, where actions are chosen according to the updated policy $\pi_{\phi_i,\theta}$.

After doing this for all tasks in $\mathbf{T}$, we continue with the meta-update step. Here, we update the parameters $\theta$ to maximise the average performance across tasks (after individually updating $\phi$ for them),

$$\theta \leftarrow \theta + \beta \nabla_\theta \frac{1}{N} \sum_{\text{MDP}_i \in \mathbf{T}} \tilde{\mathcal{J}}_{\mathcal{T}_i}(\tau_i^{\text{test}}, \pi_{\phi_i,\theta}). \tag{9}$$

This update includes higher order gradients in $\theta$ due to the dependency on (8).

### 3.3 Conditioning on Context Parameters

Since $\phi$ are independent of the network input, we need to decide where and how to condition the network on them. For an output node $h_i^{(l)}$ at a fully connected layer $l$, this can for example be done by simply concatenating $\phi$ to the inputs to that layer:

$$h_i^{(l)} = g\left(\sum_{j=1}^J \theta_{j,i}^{(l,h)} h_j^{(l-1)} + \sum_{k=1}^K \theta_{k,i}^{(l,\phi)} \phi_{0,k} + b\right), \tag{10}$$

where $g$ is a non-linear activation function, $b$ is a bias parameter, $\theta_{j,i}^{(l,h)}$ are the weights associated with layer input $h_j^{(l-1)}$, and $\theta_{k,i}^{(l,\phi)}$ are the weights associated with the context parameter $\phi_{0,k}$. This is illustrated in Figure 1. The context parameters can be added at any layer. In our experiments, we add $\phi$ at the first layer for fully connected networks.

Other conditioning methods can be used with CAML as well. E.g., for convolutional networks, we use the *feature-wise linear modulation* FiLM method (Perez et al., 2017) for image classification experiments (Section 5.2). FiLM conditions by doing an affine transformation on the feature maps: given context parameters $\phi$ and a convolutional layer that outputs $M$ feature maps $\{h_i\}_{i=1}^M$, FiLM applies a linear transformation to each feature map $FiLM(h_i) = \gamma_i h_i + \beta$, where the parameters $\gamma, \beta \in \mathbb{R}^M$ are a function of the context parameters. We use a fully connected layer $[\gamma, \beta] = \sum_{k=1}^K \theta_{k,i}^{(l,\phi)} \phi_{0,k} + b$ with the identity function at the output. In our experiments, we found it helps performance to add the context parameters not at the first layer, but after a few convolutions (in our case, after the third out of four convolution operations).

### 3.4 Context Parameter Initialisation

When learning a new task, the context parameters $\phi$ have to be initialised to some value, $\phi_0$. We argue that, instead of meta-learning this initialisation as well, a fixed $\phi_0$ is sufficient: in (10), if both $\theta_{j,i}^{(l;\phi)}$ and $\phi_0$ are meta-learned, the learned initialisation of $\phi$ can be subsumed into the bias parameter $b$, and $\phi_0$ can be set to a fixed value. The same holds for conditioning when using FiLM layers. A key benefit of CAML is therefore that it is easy to implement, since the initialisation of the context parameters does not have to be meta-learned and parameter copies are not required. We set $\phi_0 = 0$ in our implementation, which gives the additional opportunity for visual inspection of the learned context parameters (see Sections 5.1 and 5.3).

### 3.5 Learning Rate

Since the context parameters $\phi$ are inputs to the model, the gradients at this point are not backpropagated further through any other part of the model. Furthermore, because learning $\phi$ and $\theta$ is decoupled, the inner loop learning rate can effectively be meta-learned by the rest of the model. This makes the method robust to the initial learning rate that is chosen for the inner loop, as we show empirically in Sections 5.1 and 5.3.

## 4 Related Work

Meta-learning, or learning to learn, has been explored in various ways in the literature. One general approach is to learn the algorithm or update function itself (a.o., Schmidhuber (1987), Bengio et al. (1992), Andrychowicz et al. (2016), Ravi and Larochelle (2017)). Another approach is to meta-learn a model initialisation such that the model can perform well on a new task after only few gradient steps, such as MAML (Finn et al., 2017a). Other such methods are REPTILE (Nichol and Schulman, 2018) which does not require second order gradient computation and Meta-SGD (Li et al., 2017), which learns the per-parameter inner loop learning rate. Recent work (Grant et al., 2018; Finn et al., 2018) also considers Bayesian interpretations of MAML. The main difference to our work is that we consider to only adapt a small number of parameters in the inner learning loop / at test time, and that these parameters come in the form of input context parameters.

In Finn et al. (2017b) the authors augment the model with additional biases to improve the performance of MAML in a robotic manipulation setting. In contrast, we update *only* the context parameters in the inner loop, and initialise them to 0 before adaptation to a new task. A similar approach to ours, but in the context of neural language models, was done by Rei (2015). However, they do not consider the application of the method to the variety of domains covered by this paper (something briefly explored in the appendix of Finn et al. (2017a)).

Lee and Choi (2018) propose a framework to learn which parameters of the network to update in MAML, called MT-Nets. These learn a T-net and an M-Net. The M-net is a mask which decides which parameters to update in the inner loop, and is sampled (from a learned probability distribution) for each new task. The T-net is responsible for learning the task-specific update direction and step size. The idea of dynamic partitioning is attractive however it results in a more complex meta learning algorithm. In this work we consider a simpler, more interpretable alternative where the task-specific and shared parameters are disjoint sets.

Other meta-learning methods are also motivated by the fact that learning in a high-dimensional parameter space can pose practical difficulties, and fast adaptation in lower dimensional space is easier (e.g., Sæmundsson et al. (2018); Zhou et al. (2018)). Rusu et al. (2018) propose to learn a low-dimensional latent generative representation of (some of) the model parameters and perform gradient-based adaptation on a new task in this space, instead of the high-dimensional parameter space. Our method is similar but instead attempts to learn a latent representation of the *task* by backpropagating through the inner loss.

Context features as a component of inductive transfer were first introduced by Silver et al. (2008), who use a one-hot encoded task-specifying context as input to the network (which is not learned but predefined). They show that this works better than learning a shared feature extractor and having separate heads for all tasks. Learning a task embedding itself has been also explored, e.g., by Oreshkin et al. (2018), who use the task's training set to condition the network via FiLM (Perez et al., 2017) parameters. By contrast, we learn the context parameters via backpropagation through the same network that is used to solve the task.

| Method | Number of Additional Input Parameters | | | | | | |
|--------|---|------|------|------|------|------|------|
|        | 0 | 1    | 2    | 3    | 4    | 5    | 50   |
| CAML   | - | 0.84 | 0.21 | 0.20 | **0.19** | **0.19** | **0.19** |
| MAML   | 0.33 | 0.29 | 0.24 | 0.24 | 0.23 | 0.23 | 0.23 |

Table 1: MSE results of CAML and MAML for the sine curve regression task. We vary the number of input parameters, for $k = 10$ shots. Numbers are averages over $1,000$ random sets of tasks. The $95\%$ confidence intervals are $\pm 0.02$ everywhere except for CAML with 1 additional input, where it is $\pm 0.06$.

| Nodes | Parallel Partitioning | | | Stacked Partitioning | |
|-------|------------------|----------------------------------------|------------------|------------------|------------------|
|       | 1st Layer        | 1st and 2nd Layer                      | 2nd Layer        | 1st Layer        | Last Layer       |
| 1     | 3.01($\pm$0.18)  | 2.43($\pm$0.14)                        | 2.7($\pm$0.16)   | 2.39($\pm$0.14)  | 0.75($\pm$0.06)  |
| 5     | 0.29($\pm$0.02)  | 0.34($\pm$0.02)                        | 2.71($\pm$0.16)  |                  |                  |
| 20    | 0.24($\pm$0.02)  | 0.47($\pm$0.03)                        | 2.68($\pm$0.16)  |                  |                  |

Table 2: MSE results for alternative partitioning schemes on the since curve regression task. We do $k = 10$ shot learning (averaged over $1,000$ random sets of tasks, with $95\%$ confidence intervals in brackets). Labels indicate which parameters are task-specific. The rest of the network is shared across tasks and updated in the outer loop.

## 5 EXPERIMENTS

In this section we empirically evaluate CAML. Our extensive experiments aim to demonstrate three qualities of our method. First, adapting a small number of input parameters during the inner loop is sufficient to yield performance equivalent to or better than MAML in a range of regression, classification and reinforcement learning tasks. Like in MAML, it is possible to continue learning by performing several gradient update steps at test time, even when training using only one gradient step. Second, CAML is robust to the task-specific learning rate and scales well to more expressive networks without overfitting. Third, an embedding of the task emerges in the context parameters solely via backpropagation through the original inner loss. The implementation and scripts to reproduce the results are available at [blinded for review].

### 5.1 REGRESSION

We start with the regression problem of fitting sine curves, using the same setup as Finn et al. (2017a) to allow a direct comparison. A task is defined by the amplitude and phase of the sine curve, and is generated by uniformly sampling the amplitude from $[0.1, 0.5]$ and the phase from $[0, \pi]$. For training, ten labelled datapoints (uniformly sampled from $x \in [-5, 5]$) are given for each task for the inner loop update. Per meta-update we iterate over a batch of 25 tasks and perform gradient descent on a mean-squared error (MSE) loss. We use a neural network with two hidden layers and 40 nodes each and ReLU non-linearities. During testing we present the model with ten datapoints from 1000 newly sampled tasks and measure MSE over 100 test points.

CAML uses the same training procedure and architecture but adds context parameters. To allow a fair comparison, we add the same number of additional inputs to MAML, an extension that was also done by Finn et al. (2017b). These additional parameters are meta-learned together with the rest of the network, which can improve performance due to a more expressive gradient. Our method differs from this formulation in that we update only the context parameters in the inner loop, and reinitialise them to zero for each new task. In the outer loop, we only update the shared parameters.

Table 1 shows that CAML outperforms the original MAML (with no additional inputs) significantly, and MAML with the same network architecture by a small margin. This performance gain is possible even though at test time, CAML adapts only 2-5 parameters, instead of around 1600. To test the hypothesis that it suffices to adapt only *input parameters* per task, we also compare to alternative parameter partitions in Table 2. In *parallel partitioning*, we choose a strict *subset* of the nodes of each layer for task-specific adaptation, and meta-learn the rest. In *stacked partitioning*, we choose one or several *layers* for task-specific adaptation, and meta-learn the other layers. The results confirm that partitioning on context parameters is key to success: the other variants perform worse, often significantly so. A recent method was proposed by Lee and Choi (2018), where they also partition the network to adapt only part of it on a specific task – the partitioning mask, however, is learned. They

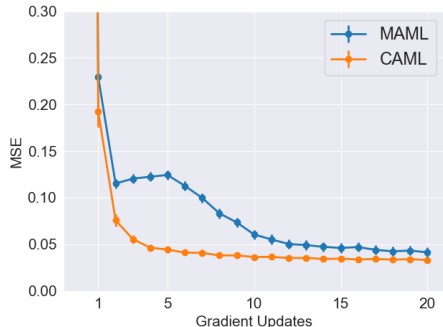

Figure 2: Performance after several gradient steps (on the same batch) averaged over 1000 unseen tasks. The size of the context parameter / additional input to MAML is 5.

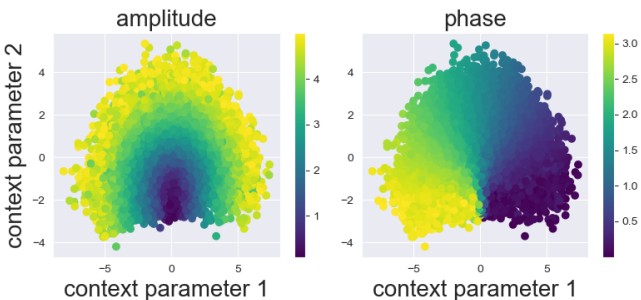

Figure 3: Visualisation of what the context parameters learn given a new task. In this case we have 2 context parameters, and shown is the value they take after 5 gradient update steps on a new task. Each dot is one random task, with its colour indicating the amplitude (left) or phase (right) of that task.

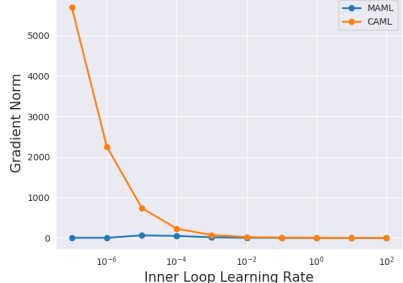

Figure 4: CAML scales the model weights so that the inner learning rate is compensated by the context parameters gradients magnitude.

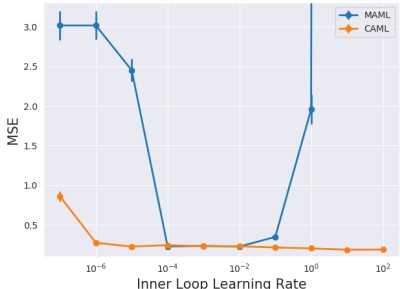

Figure 5: Measuring performance for different learning rates shows that CAML is more robust to this hyperparameter than MAML.

test their method on the regression task as well, but we outperform the numbers they report significantly (not shown, since we believe this might be due to implementational details). In the next section we will see that on few-shot classification, this approach achieves comparable performance to our method.

MAML is known to keep learning after several gradient update steps. We test this on our method as well, with the results shown in Figure 2 for up to 10 gradient steps. CAML outperforms MAML even after taking several gradient update steps, and is more stable, as indicated by the size of the confidence intervals and the monotonic learning curve.

As described in Section 3.4, CAML has the freedom to scale the gradients at the context parameters since they are inputs to the model and trained separately. Figure 4 plots the inner learning rate against the norm of the gradient of the context parameters at test time. We can see that the weights are adjusted so that lower learning rates bring about larger context parameter gradients and vice-versa. This results in the method being extremely robust to learning rates as confirmed by Figure 5. We plot the performance while varying the learning rate from $10^{-7}$ to $10^2$. CAML is robust to changes in learning rate while MAML performs well only in a small range. Work by Li et al. (2017) shows that MAML can be improved by learning a parameter-specific learning rate, which, however, introduces a lot of additional parameters.

CAML's performance on the regression task correlates with how many variables are needed to encode the tasks. In these experiments, two parameters vary between tasks, which is exactly the context parameter dimensionality at which CAML starts to perform well (the optimal encoding is three dimensional, as phase is periodic). This suggests CAML may indeed learn task descriptions in the context parameters. Figure 3 illustrates this by plotting the value of the learned inputs against the amplitude/phase of the task in the case of two context parameters. The model learns a smooth embedding in which interpolation between tasks is possible.

| Model | 5-way accuracy | |
| --- | --- | --- |
| | 1-shot | 5-shot |
| Matching Nets (Vinyals et al., 2016) | 46.6% | 60.0% |
| Meta LSTM (Ravi and Larochelle, 2017) | $43.44 \pm 0.77\%$ | $60.60 \pm 0.71\%$ |
| Prototypical Networks (Snell et al., 2017) | $49.42 \pm 0.78\%$ | $\mathbf{68.20} \pm 0.66\%$ |
| Meta-SGD (Li et al., 2017) | $50.47 \pm 1.87\%$ | $64.03 \pm 0.94\%$ |
| REPTILE (Nichol and Schulman, 2018) | $49.97 \pm 0.32\%$ | $65.99 \pm 0.58\%$ |
| PLATIPUS (Finn et al., 2018) | $50.13 \pm 1.86\%$ | |
| MT-NET (Lee and Choi, 2018) | $\mathbf{51.70} \pm 1.84\%$ | |
| VERSA (Gordon et al., 2018) | $50.70 \pm 0.86\%$ | $65.03 \pm 0.66\%$ |
| MAML (32) (Finn et al., 2017a) | $48.07 \pm 1.75\%$ | $63.15 \pm 0.91\%$ |
| MAML (64) | $44.70 \pm 1.69\%$ | $61.87 \pm 0.93\%$ |
| CAML (32) | $47.24 \pm 0.65\%$ | $59.05 \pm 0.54\%$ |
| CAML (64) | $49.56 \pm 0.68\%$ | $63.94 \pm 0.55\%$ |
| CAML (128) | $49.84 \pm 0.68\%$ | $64.63 \pm 0.54\%$ |
| CAML (256) | $51.23 \pm 0.70\%$ | $65.20 \pm 0.54\%$ |
| CAML (512) | $\mathbf{51.82} \pm 0.65\%$ | $65.85 \pm 0.55\%$ |
| CAML (256, first order) | $49.84 \pm 0.71\%$ | $64.26 \pm 0.55\%$ |
| CAML (512, first order) | $49.92 \pm 0.68\%$ | $63.59 \pm 0.57\%$ |

Table 3: Few-shot classification results on the Mini-Imagenet test set (average accuracy with 95% confidence intervals on a random set of 1000 tasks). We compare to existing methods on this benchmark that use deep convolutional networks, and MAML with a larger network (results obtained with the author's open sourced code, with all hyperparameters unchanged except the number of filters).

## 5.2 CLASSIFICATION

To evaluate CAML on a more challenging problem, we test it on the competitive few-shot image classification benchmark Mini-Imagenet (Ravi and Larochelle, 2017). In $N$-way $K$-shot classification. a task is a random selection of $N$ classes, for each of which the model gets to see $K$ examples. From these it must learn to classify unseen images from the $N$ classes. The Mini-Imagenet dataset consists of 64 training classes, 12 validation classes, and 24 test classes. During training, we generate a task by selecting $N$ classes at random from the 64 classes and training the model on $K$ examples of each, i.e., a batch of $N \times K$ images. The meta-update is done on a set of unseen images of the same classes.

On this benchmark, MAML uses a network with four convolutional layers with 32 filters each and one fully connected layer at the output (Finn et al., 2017a). We use the same network architecture, but with between 32 and 512 filters per layer. We use 100 context parameters and add a FiLM layer (see Section 3.3) that conditions on these after the third convolutional layer. The parameters of the FiLM layer are meta-learned with the rest of the network, i.e., they are part of $\theta$. All our models were trained with two gradient steps in the inner loop and evaluated with two gradient steps (note: MAML was trained with five inner-loop gradient steps and evaluated with ten gradient steps). The inner learning rate was set to $1.0$. Following Finn et al. (2017a), we ran each experiment for $60K$ meta-iterations and selected the model with the highest validation accuracy for evaluation on the test set.

Table 3 shows our results on Mini-Imagenet held-out test data for 5-way 1-shot and 5-shot classification. We compare to a number of existing meta-learning approaches that use convolutional neural networks, including MAML. Our largest model (512 filters) clearly outperforms MAML, and outperforms the other methods on the 1-shot classification task. On 5-shot classification, the best results are obtained by prototypical networks (Snell et al., 2017), a method that is specific to few-shot classification and works by computing distances to prototype representations of each class. Our smallest model (32 filters) under-performs MAML (within the confidence intervals). As we can see, CAML benefits from increasing model expressiveness: since we only adapt the context parameters in the inner loop per task, CAML can substantially increase the network size, without overfitting during the inner loop update. We tested scaling up MAML to a larger network size as well (see Table 3), but found that this hurt accuracy. The table only compares CAML to approaches that use conventional convolutional neural networks. Approaches that use much deeper, residual networks (e.g., Gidaris and Komodakis (2018), (Bauer et al., 2017), (Oreshkin et al., 2018), (Qiao et al., 2017)) can achieve higher accuracies. To the best of our knowledge, the LEO method by Rusu et al. (2018) is the current state of the art

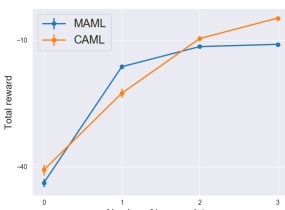 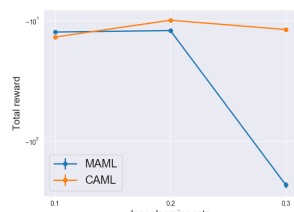 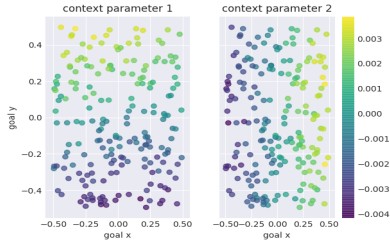

(a) Performance per gradient update.  (b) Performance per learning rate.  (c) Learned task embedding.

Figure 6: 2D navigation task analysis. Figure 6a shows the performance of each method as more gradient updates are performed. Figure 6b shows the performance of each method after 2 updates as the inner loop learning rate is increased. As in the case of regression CAML is not afected by this parameter. Figure 6c describes the goal position of different 2D navigation tasks and the corresponding context parameter activation obtained by performing 2 gradient updates. We can see that the context parameters represent an interpretable embedding of the task at hand. Context parameter 1 seems to encode the y position, while context parameter 2 encodes the x position.

with 60% and 75.7% accuracy on 1 and 5-shot respectively. Our method can be readily applied to deep residual networks as well, and we leave this exploration for future work. Table 3 also shows the first order approximation of our largest models, where the gradient with respect to $\theta$ is not backpropagated through the inner loop update of the context parameters $\phi$. As expected, this results in a lower accuracy (a drop of $1 - 2\%$), but we are still able to outperform MAML with a first-order version of our largest network.

Thus, CAML can achieve much higher accuracies than MAML by increasing the network size, without overfitting. Our results are obtained by only adjusting 100 parameters at test time, instead of $> 30,000$ like MAML.

## 5.3 REINFORCEMENT LEARNING

To demonstrate the versatility of CAML, we also perform preliminary reinforcement learning experiments on a 2D Navigation task, also introduced by Finn et al. (2017a). In this domain, the agent moves in a 2D world using continuous actions. At each timestep it is given a negative reward proportional to its distance from a pre-defined goal position. Each task is defined by a new unknown goal position.

We follow the same procedure as Finn et al. (2017a). Goals are sampled from an interval of $(x, y) = [-0.5, 0.5]$. At each step we sample 20 tasks for both the inner and outer loops and testing is performed on 40 new unseen tasks. We perform learning for 500 iterations and the best performing policy during training is then presented with new test tasks and allowed two gradient updates. For each update, the total reward over 20 rollouts per task is measured. We use a two-layer network with 100 units per layer and ReLU non-linearities to represent the policy and a linear value function approximator. For CAML we use five context parameters at the input layer.

In terms of performance (Fgure 6a) we can see that the two methods are highly competitive. MAML performs better after the first gradient update after it is surpassed by CAML. Figure 6b, which plots performance for several learning rates, shows that CAML is again less sensitive to the inner loop learning rate. Only when using a learning rate of 0.1 is MAML competitive in performance. Furthermore, CAML adapts 5 parameters whereas MAML adapts around 10,000 parameters.

As with regression, the optimal task embedding is low dimensional enough to plot. We therefore apply CAML with two context parameters and plot how these correlate with the actual position of the goal for 200 test tasks. Figure 6c shows that the context parameters obtained after two policy gradient updates represent a disentangled embedding of the actual task. Specifically, context parameter 1 appears to encode the $y$ position of the goal, while context parameter 2 encodes the $x$ position. Hence, CAML can learn compact potentially interpretable task embeddings via backpropagation through the inner loss.

## 6 CONCLUSION AND FUTURE WORK

In this paper we introduced CAML, a meta-learning approach for fast adaptation that introduces context parameters in addition to the model's parameters. The context parameters are used to modulate the whole network during the inner loop of meta-learning, while the rest of the network parameters are adapted in the outer loop and shared across tasks. On regression, our method outperforms MAML and is superior to naive approaches to partitioning network parameters. We also showed that CAML is highly competitive with state of the art methods on few shot classification using CNNs. In addition to this, we experimented extensively with some unique properties that specifically arise from the way that our method is formulated, such as robustness to learning rate and the emergence of task embeddings at the context parameters. Another interesting extension would be to inspect the context parameter representations learned by CAML on the Mini-Imagenet benchmark using advanced dimensionality reduction techniques.

In this paper we performed some preliminary RL experiments. We are interested in extending CAML to more challenging problems and explore its role in allowing for smart exploration in order to identify the task at hand. It would also be interesting to consider probabilistic extensions along the lines of PLATIPUS (Finn et al., 2018) where the context parameters include uncertainty about the task.

Finally, the intriguing empirical properties of CAML detailed in this work will be the base of more theoretical investigations in the future.

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

# Appendix

## A   PSEUDO-CODE

---
**Algorithm 1** CAML for Supervised Learning

---
**Require:** Distribution over tasks $p(\mathcal{T})$
**Require:** Step sizes $\alpha$ and $\beta$
**Require:** Initial model $f_{\phi_0,\theta}$ with $\theta$ initialised randomly and $\phi_0 = 0$
 1: **while** not done **do**
 2:     Sample batch of tasks $\mathbf{T} = \{\mathcal{T}_i\}_{i=1}^N$ where $\mathcal{T}_i \sim p$
 3:     **for all** $\mathcal{T}_i \in \mathbf{T}$ **do**
 4:         $\mathcal{D}_i^{\text{train}}, \mathcal{D}_i^{\text{test}} \sim q_{\mathcal{T}_i}$
 5:         $\phi_0 = 0$
 6:         $\phi_i = \phi_0 - \alpha \nabla_\phi \frac{1}{M_i^{\text{train}}} \sum\limits_{(x,y) \in \mathcal{D}_i^{\text{train}}} \mathcal{L}_{\mathcal{T}_i}(f_{\phi_0,\theta}(x), y)$
 7:     **end for**
 8:     $\theta \leftarrow \theta - \beta \nabla_\theta \frac{1}{N} \sum\limits_{\mathcal{T}_i \in \mathbf{T}} \frac{1}{M_i^{\text{test}}} \sum\limits_{(x,y) \in \mathcal{D}_i^{\text{test}}} \mathcal{L}_{\mathcal{T}_i}(f_{\phi_i,\theta}(x,y))$
 9: **end while**

---

---
**Algorithm 2** CAML for RL

---
**Require:** Distribution over tasks $p(\mathcal{T})$
**Require:** Step sizes $\alpha$ and $\beta$
**Require:** Initial policy $\pi_{\phi_0,\theta}$ with $\theta$ initialised randomly and $\phi_0 = 0$
 1: **while** not done **do**
 2:     Sample batch of tasks $\mathbf{T} = \{\mathcal{T}_i\}_{i=1}^N$ where $\mathcal{T}_i \sim p$
 3:     **for all** $\mathcal{T}_i \in \mathbf{T}$ **do**
 4:         Collect rollout $\tau_i^{\text{train}}$ using $\pi_{\phi_0,\theta}$
 5:         $\phi_i = \phi_0 + \alpha \nabla_\phi \tilde{\mathcal{J}}_{\mathcal{T}_i}(\tau_i^{\text{train}}, \pi_{\phi_0,\theta})$
 6:         Collect rollout $\tau_i^{\text{test}}$ using $\pi_{\phi_i,\theta}$
 7:     **end for**
 8:     $\theta \leftarrow \theta + \beta \nabla_\theta \frac{1}{N} \sum\limits_{\mathcal{T}_i \in \mathbf{T}} \tilde{\mathcal{J}}_{\mathcal{T}_i}(\tau_i^{\text{test}}, \pi_{\phi_i,\theta})$
 9: **end while**

---

## B   EXPERIMENTS

### B.1   CLASSIFICATION

For Mini-Imagenet, our model takes as input images of size $84 \times 84 \times 3$ and has 5 outputs, one for each class. The model has four modules that each consist of: a $2D$ convolution with a $3 \times 3$ kernel, padding 1 and 128 filters, a batch normalisation layer, a max-pooling operation with kernel size 2, if applicable a FiLM transformation (only at the third convolution, details below), and a ReLU activation function. The output size of these four blocks is $5 \times 5 \times 128$, which we flatten to a vector and feed into one fully connected layer.

The FiLM layer itself is a fully connected layer with inputs $\phi$ and a 256-dimensional output and the identity function at the output. The output is divided into $\gamma$ and $\beta$, each of dimension 128, which are used to transform the filters that the convolutional operation outputs. The context vector is of size 100 (other sizes tested: 50, 200) and is added after the third convolution (other versions tested: at the first, second or fourth convolution).

The network is initialised using He et al. (2015) initialisation for the weights of the convolutional and fully connected weights (including the FiLM layer weights). The bias parameters are initialised to zero, except at the FiLM layer.

We use the Adam optimiser for the meta-update step with an initial learning rate of $0.001$. This learning rate is annealed every $5,000$ steps by multiplying it by $0.9$. The inner learning rate is set to $0.1$ (other hyperparameters tested: $1.0, 0.01$).

For Mini-Imagenet, we use a meta batchsize of $4$ and $2$ tasks for 1-shot and 5-shot classification respectively. For the batch norm statistics, we always use the current batch – also during testing. I.e., for 5-way 1-shot classification the batch size at test time is $5$, and we use this batch for normalisation.

## C PRACTICAL TIPS

### C.1 IMPLEMENTATION

The context parameters $\phi$ can be added to any network, and do not require direct access to the rest of the network weights like MAML. In PyTorch this can be done as follows. To add CAML parameters to a network, it is necessary to first initialise them to zero when the model is initialised:

```
self.context_params = torch.zeros(size=[self.num_context_params],
requires_grad=True)
```

Add a way to reset the context parameters to zero (e.g., a method that just does the above). During the forward pass, add the context parameters to the input by concatenating it (when using a fully connected network):

```
x = torch.cat((x, self.context_params.expand(x.shape[0], -1)), dim=1)
```

(This is for fully connected networks. We refer the reader to our implementation for how to use FiLM to condition CNNs.) To correctly set the computation graph for the outer loop, it is necessary to assign the context parameters manually with their gradient. In the inner loop, compute the gradient:

```
grad = torch.autograd.grad(task_loss, model.context_params,
create_graph=True)[0]
```

The option *create_graph* will make sure that you can take the gradient of *grad* again. Then, update the context parameters using one gradient descent step

```
model.context_params = model.context_params - lr_inner * grad
```

If you now do another forward pass and compute the gradient of the model parameters $\theta$ (for the outer loop), these will include higher order gradients because *grad* above includes gradients of $\theta$, and because we kept the computation graph via the option *grad*. To see how to train CAML and aggregate the meta-gradient over several tasks, see our implementation at [blinded for review].

### C.2 HYPERPARAMETER SELECTION

The choice of network architecture/size and context parameters can be guided by domain knowledge. E.g., for the few-shot image classification problem, an appropriate model is a deep convolutional model. For the context parameters, it is important to make sure they are not underparameterised. CAML can deal with larger than necessary context parameters (see Table 1), however, at some point it would probably overfit to the current task in the inner loop. We have not experienced this in practise though. Regarding learning rates, we suggest to start with an inner loop learning rate of 1 and the Adam optimiser (Kingma and Ba, 2014) with the standard learning rate of $0.001$ for the outer loop

For CNNs, we found that adding the context parameters not at the input layer, but after several (in our case after the third out of four) convolutions works best. We believe this is because the lower-level features that the first convolutions extract are useful for any image classification task, and we only want our task embedding to influence the activations at the deeper layers. In our experiments we used a FiLM network with no hidden layers. We tried deeper versions, but this resulted in inferior performance.

We also tested to add context parameters at several layers instead of only one. However, in our experience this resulted in similar (regression and RL) or worse (in the case of CNNs) performance.

