# OpenReview forum: "CAML: Fast Context Adaptation via Meta-Learning"
_ICLR.cc/2019/Conference_

### Official Review · AnonReviewer3 · 2018-11-02
**An interesting meta-learning algorithm**

**Rating:** 6
**Confidence:** 2

**Review:**

CAML seems an interesting meta-learning algorithm. I like the idea that the context parameters are used to modulate the whole network during the inner loop of meta-learning, while the rest of the network parameters are adaped in the outer loopand shared across tasks. Also, it is good to see that CAML is competitive with on few shot CNNs.

The paper is very well presented. Experiments are reasonably solid.

If I understood correctly, although CAML has achieved better accuracy it seems CAML still requires a decent amount of parameter/network structure optimisation. Would be good if the paper has a section talking about practical tricks of how to find the best CAML hyperparameter quickly.

---

> ### Author Response · Authors · 2018-11-18
> **Reply**
>
> Thank you for your review, and the time to assess our paper.
>
> We added a section with practical tips for implementation and hyperparameter selection to the Appendix. In general, we think choosing the hyperparameters in CAML can be guided by domain knowledge: since we separate the task-specific and shared parameters, the choice of both is more intuitive for the human designer than in MAML. The context parameters of CAML can be added on top of any network architecture, and they are updated only via backpropagation (unlike, e.g., LEO [1] which requires a separate network to encode the training data). Additionally, our method is not sensitive to network architecture (not prone to overfit like MAML), the inner loop learning rate, and can handle overparameterisation of the context parameters (as shown in the regression experiment, see updated Table 1).
>
> [1] “Meta-Learning with Latent Embedding Optimisation” Rusu et al. (2018)

---

### Official Review · AnonReviewer2 · 2018-11-02
**Good paper in general**

**Rating:** 6
**Confidence:** 2

**Review:**

They are proposing a meta-learning method inspired by previous method, MAML. Their idea is separating the parameters in to two groups of context and shared parameters. The context parameters are learned through back-propagation of inner-loop and represents embedding for individual task. Shared-parameters on the other hand are shared between all tasks, and are learned in the outer-loop.

Compared to MAML, the pros of their method is as follows:
- Less sensitive to learning rate: thus more robust to hyper parameters.
- Does not prone to overfit as MAML does.
- It is easier to implement, more efficient from memory view point.

Cons in general,
-  In Mini-ImgeNet data set, although they are beating MAML, but they are not able to beat other competitors in 5-shot classification.
- They could have explored applying their method to deep residual networks and compare their results.

---

> ### Author Response · Authors · 2018-11-18
> **Reply**
>
> Thank you for your time to read and evaluate our paper.

---

### Official Review · AnonReviewer1 · 2018-11-07
**interesting idea, falling short on experimental evidence**

**Rating:** 6
**Confidence:** 4

**Review:**

The paper talks about Meta-Learning where some of the parameters of the models adapt to the new task (context parameters) and rest of the parameters are kept fixed (shared parameters). The authors propose a more general approach and show how CAML works for supervised learning and reinforcement learning paradigms.

quality - The paper is written with good mathematical notation and in general is of high quality. The references to related work and motivation of the problem is good.

clarity - While the paper is clear in many parts, it can be a lot better. Specifically it is unclear why authors chose regression, classification and RL to make their point without landing either one of them fully confidently.

originality - the idea is good and general enough to be applicable for many situations. While variants of this idea have been tried with fine-tuning for transferred learning I still think this work can classify as original and novel.

significance of this work - The significance of meta learning is good but based on the experiments authors conducted I am worried it has little significance.

pros and cons - Overall, while I am supportive of a weak accept because of the idea and it's broad applicability I feel authors should maybe chose one of the tasks and show much more value in using the CAML framework. The three tasks they chose are all toy problems and do not instill confidence in the validity of CAML for either large scale experiments or in setups where distribution is changing but tasks remain same. It would be great to strengthen the paper with a more cleaner story on the experiments section and show CAML achieves SOA convincingly.

---

> ### Author Response · Authors · 2018-11-18
> **Reply**
>
> Thank you for your time and review of our paper.

---

### Official Review · AnonReviewer4 · 2018-11-09
**incremental idea, weak experimental evidence**

**Rating:** 4
**Confidence:** 5

**Review:**

Summary
CAML is a gradient-based meta-learning method closely related to MAML. It divides model parameters into disjoint sets of task-specific parameters $\phi$ which are adapted to each task and task-independent parameters $\theta$ with are meta-learned across tasks. $\phi$ are then interpreted as an embedding and fed as input to the model (parameterized by $\theta$). Experiments demonstrate that this approach performs on par with MAML while adapting far fewer parameters. An additional benefit is that this approach is less sensitive to the adaptation learning rate and is easier to implement and faster to compute.

Strengths
While not really explained in the paper, this work connects gradient-based to embedding-based meta-learning approaches. Adaptation is via gradient descent, but the adapted parameters are then re-interpreted as an embedding.
The method has the potential to perform on par with MAML while being simpler and faster.
The paper is well-written.

Weaknesses
The field of meta-learning variants is crowded, and this paper struggles to carve out its novelty.
Rusu et al (LEO) optimize a context vector, which is used to generate model parameters. Reducing the generative model to a point estimate, how is this different from generating the FiLM parameters as a function of context as done in CAML?
Lee and Choi (MT-nets) propose a general formulation for learning which model parameters to adapt. CAML is simpler in that the model parameters to adapt are chosen beforehand to be inputs.
Snell et al. / Oreshkin et al. are prototype-based methods infer context via a neural network rather than optimizing for it.

In this context, CAML appears to be yet another point drawn from the convex hull of choices already explored in episodic meta-learning (these choices can be broadly grouped into task encoding and conditional inference). The paper must then rest on its experimental results, which are at present unconvincing.

On the whole, the experimental results seem weak and analysis results largely uninformative. The method is benchmarked on the toy tasks of sinusoid regression and a 2-D point mass, as well as mini-ImageNet few-shot classification. The sinusoid and point mass navigation are toy and compared only to MAML, so it is hard to draw conclusions from those experiments. For mini-ImageNet, while CAML outperforms MAML, it seems that the pertinent comparison is with MT-NET (which CAML does not outperform) and LEO (missing fair comparison?).

Questions regarding experiments
 - CAML is robust to the adaptation learning rate, but isn’t this true of any scheme that separates meta-learned and adapted parameters into disjoint sets? (e.g. also true of Lee and Choi?)
 - The visualizations of the context parameters are nice, but interpreting much higher dimensional context vectors (which would be necessary for harder tasks) is more difficult, so I’m not sure what to take away from this? It’s very unsurprising that the 2-D context vector encodes x and y position in the point mass experiment, for example.
 - I am confused by the comparison between adapting input parameters versus subsets of nodes at each layer or entire layers for the sinusoid regression task. Adapting subsets of nodes at each layer roughly corresponds to Lee and Choi, yet the reported numbers are quite different?
 - In Table 3, which CAML is a fair comparison (in terms of network size and architecture) to MT-NET?

Editorial Notes
Intro paragraph 3: fine-tuning image classification features for a semantic segmentation task is not a good example of task independent parameters, since fine-tuning end-to-end gives significant improvements.
Related work paragraph 2: Initializing context parameters to zero is not the only difference with Rei et al (2015), and seems a strange thing to highlight?
Tables 1 and 2: state what the task is in the caption

---

> ### Author Response · Authors · 2018-11-18
> **Reply (Part 1)**
>
> Thank you for the time to evaluate our paper, and the thorough review. We address your raised points below.
>
> “Rusu et al (LEO) optimize a context vector, which is used to generate model parameters. Reducing the generative model to a point estimate, how is this different from generating the FiLM parameters as a function of context as done in CAML?”
> - The outputs of the FiLM layer can be seen as parameters of the network, but this differs from the approach in LEO as follows. The FiLM outputs scale and shift entire feature maps in convolutional layers (but have no influence on the FiLM layer parameters, or convolution parameters, themselves). LEO generates the weights of the last layer of the neural network (for classification) or the entire parameter vector of the network (in the general case). We view the context parameter in CAML as modulating the activations in a fixed network, whereas LEO generates the parameters themselves.
>
> “Lee and Choi (MT-nets) propose a general formulation for learning which model parameters to adapt. CAML is simpler in that the model parameters to adapt are chosen beforehand to be inputs.”
> - Lee and Choi learn which parameters to adapt, but they do not consider having additional inputs that can be adapted. Additionally, MT-Nets do not partition the network parameters into disjoint sets (a new mask is drawn for each new task, from a learned probability distribution; all parameters are updated in the outer loop). We introduce context parameters and show that this is sufficient compared to the more complex approach of MT-Nets, and that by this we can learn a task embedding via backpropagation.
>
> “Snell et al. / Oreshkin et al. are prototype-based methods infer context via a neural network rather than optimizing for it.”
> - Both these methods are specific to few-shot classification, whereas CAML can also be applied to regression and reinforcement learning. Methods for few-shot classification often rely on learning class embeddings, whereas we directly learn an embedding for the current task (i.e., all classes) which modulates the classification network.
>
> “CAML is robust to the adaptation learning rate, but isn’t this true of any scheme that separates meta-learned and adapted parameters into disjoint sets? (e.g. also true of Lee and Choi?) “
> - We don’t think that’s necessarily true. If the task-specific parameters depend on shared parameters in earlier layers of the network, regulating the learning rate via the magnitude of the gradient would have an influence on the outer loop update (since those gradients would be backpropagated further). This can be countered to some extent, but might still be less flexible than CAML, where the context parameters are leaves of the computation graph.
> MT-Nets (Lee and Choi) learn an M and a T net. The M-net is responsible for selecting which parameters to update, and is sampled for each new task (from a learned probability distribution). If we understand correctly, all parameters are updated in the outer loop - hence the parameter sets are not entirely disjoint. The T-net learns the update direction and step size of the parameters, which is why MT-Nets are robust to the inner loop learning rate. For the regression task we show that CAML is robust within a learning rate range of 10^(-7) to 100 (see updated Figure 5), whereas MT-Nets cannot successfully scale to a learning rate of 10 (see their paper).
>
> “The visualizations of the context parameters are nice, but interpreting much higher dimensional context vectors (which would be necessary for harder tasks) is more difficult, so I’m not sure what to take away from this?”
> - We show the visualisations in the 2-D context to confirm that we indeed learn a task embedding via backpropagation, which corresponds to (and is smooth with respect to) the true task differences. This illustrates that the inner loop of meta-learning algorithms can be seen as a task embedding step, and that we successfully do this via backpropagation. For higher dimensional context vectors, visualisation methods such as t-SNE could be used. Few-shot classification is a special case, since the embedding is not disentangled (with respect to the different classes). This makes visualisation for separate classes difficult. If a disentangled representation is of interest, it might be possible to train CAML with this in mind, e.g. by updating only a part of the context vector per class.
>
> [continued below]

---

> > ### Author Response · Authors · 2018-11-18
> > **Reply (Part 2)**
> >
> > “I am confused by the comparison between adapting input parameters versus subsets of nodes at each layer or entire layers for the sinusoid regression task. Adapting subsets of nodes at each layer roughly corresponds to Lee and Choi, yet the reported numbers are quite different?”
> > - Yes, adapting subsets of nodes at each layer corresponds roughly to Lee an Choi, except that we choose which subset of nodes are adapted for this ablation study of alternative partitioning schemes. The MSE reported by Lee and Choi is higher than ours, and also their MSE scores for MAML are higher than in the original paper. We assume that this is due to differences in the implementation.
> >
> > “In Table 3, which CAML is a fair comparison (in terms of network size and architecture) to MT-NET?”
> > - MT-Nets use the same architecture as MAML (32 filters), so the expressiveness during the forward pass is the same as our smallest (32 filter) architecture. MT-Net additionally learns parameters to generate T and M, which for this network are around 4,000 parameters. To outperform MT-Nets, we need to scale up the number of filters in our convolutions - trading off implementation complexity (higher in MT-Nets) for a larger network (necessary for CAML) and having a separate task embedding (given in CAML).

---

> > > ### Comment · AnonReviewer4 · 2018-11-30
> > > **Thank you for clarifications, still have concerns**
> > >
> > > Thank you for the detailed replies, particularly regarding how CAML relates to prior work.
> > >
> > > I still have concerns about novelty and strength of experiments. Rusu et al. learn an embedding that can also be interpreted as a task encoding, and it’s not clear from the results whether the choice of parameter regression (LEO) versus feature fusion (CAML) matters much. While CAML is admirably simpler, the experiments don’t convincingly make the case that this simple change gives significant benefits.
> > > Experiments on more complex few-shot learning problems might illuminate these benefits.

---

### Public Comment · ~Ali_Janalizadeh_Choobbasti1 · 2018-10-30
**Making the model more scalable**

A very nice read, the work is very admirable.

I was thinking if there was a way to split the context parameters used in learning the policy into two separate streams; a linear and nonlinear stream. Something like in nonlinear control theory, where the linear stream would stabilize the local dynamics, and the nonlinear stream would handle global control.
In a reinforcement learning setup, this would bring the benefits of both linear and nonlinear policies, which would, in turn, lead to greater generalization and more scalability.

---

> ### Author Response · Authors · 2018-11-01
> **Two-Stream Architecture**
>
>
> Thank you for the kind feedback and your comment.
>
> Yes, splitting up the context parameters and the network architecture up into separate streams like that is possible with CAML: given the two forward streams, there would also be two backward streams through which the gradient gets propagated, for the respective parts of the context parameters. It would be interesting to see the network can make use of the opportunity to propagate information through the separate streams to speed learning.

---

### Author Response · Authors · 2018-11-26
**Rebuttal**


We thank the reviewers for their time to evaluate our paper, and their valuable feedback.

We uploaded a revision of the paper. Besides editorial changes, we added a section of practical tips to the Appendix as suggested by Reviewer 3. We updated our related work section to better reflect how CAML differs from existing work, in particular MT-Nets [1] and LEO [2], in response to Reviewer 4.

We summarise these differences here:

MT-Nets [1] learn which parameters to update in MAML [3]. To this end, they learn an M-net, which is a mask (sampled from a learned probability distribution for each new task), and determines which parameters are updated in the inner loop. In the outer loop, all parameters are updated. Hence the task-specific and shared-task parameters are not disjoint as in CAML, and no task embedding emerges. Additionally, they learn a T-net, which learns the update direction and step size of the parameters, which makes MT-Nets robust to the inner loop learning rate. CAML adjusts the inner loop learning rate automatically via the magnitude of the gradient, and can handle a wider range of initial learning rates compared to MT-Nets. This is possible because the parameter sets are disjoint, and the context parameters are inputs to the model (i.e., gradients do not get backpropagated further).

LEO [2] learns an embedding which generates the weights of the last layer (for classification) or the entire parameter vector of the network (in the general case). This embedding is computed via an embedding network and a relation network. At test time, the gradient steps are done in this embedding space (with additional fine-tuning of the generated network for the Mini-Imagenet experiments). In contrast, we learn an embedding that modulates a fixed network, and we do so via backpropagation through that same network. Hence our method has fewer hyperparameters / architecture choices that have to be made.

Some reviewers raised concerns about the experimental evaluation of CAML. We deliberately chose to show that CAML works well on a broad range of problems, instead of focusing on a single setting.

CAML can be scaled up to achieve better performance on the Mini-Imagenet benchmark, but we see this as an orthogonal problem and a question of compute and hyperparameter search. After the reviewers’ feedback, we ran an additional experiment using a Resnet-18 model to test how feasible it is to scale CAML up. We tested the same hyperparameters that we used for the CNN-based experiments. The implementation is easy: we used the ResNet readily available in PyTorch and added the context parameters / FiLM layer in-between the second and third residual block, together with a few lines of extra functionality (like resetting the context parameters to zero). For MAML / MT-Nest, we would have to manually access all network parameters to set up the computation graph. We get 52.16% (+/- 0.32) and 66.33% (+/- 0.26) accuracy on the 1 and 5 shot problem respectively, which is higher than our best CNN-based results in the paper and outperforms MT-Nets. Again, these results are achieved by adjusting only 100 context parameters at test time. This indicates that CAML can indeed be scaled up further, and we leave it to future work to try larger resnets and do a full hyperparameter search. (Note that these numbers might not be directly comparable to the SOTA scores of LEO [2], who get their final score by training on the training and validation set.)

Overall, we believe our paper is interesting to the ICLR community, since compared to the popular algorithm MAML [3] and several papers that build on it (like Meta-SGD [4] and MT-Net [1]s) we explicitly learn task embeddings, which are separated from the network that is shared across tasks. We interpret the inner loop of meta-learning as a task identification step, and show that we only need to adapt a few parameters at test time, instead of the entire network. Our paper is the first to show that this is possible for a wide range of problems using a simple backpropagation operation on contextual input parameters. Compared to MAML, our method has the advantage of being robust to overfitting. It is also easier to implement (we do not need to access all weights of the network manually), needs fewer memory writes, and can be useful for distributed machine learning systems.

[1] “Gradient-based meta-learning with learned layerwise metric and subspace” Lee et al. (2018)
[2] “Meta-Learning with Latent Embedding Optimisation” Rusu et al. (2018)
[3] “Model-Agnostic Meta-Learning for Fast Adaptation of Deep Networks” Finn et al. (2017)
[4] “Meta-SGD: Learning to learn quickly for few shot learning” Li et al. (2017)

---

### Meta-Review · Area_Chair1 · 2018-12-14

**Confidence:** 4
**Recommendation:** Reject

**Metareview:**

This paper proposes a meta-learning algorithm that performs gradient-based adaptation (similar to MAML) on a lower dimensional embedding. The paper is generally well-written, and the reviewers generally agree that it has nice conceptual properties. The method also draws similarities to LEO. The main weakness of the paper is with regard to the strength of the experimental results. In a future version of the paper, we encourage the authors to improve the paper by introducing more complex domains or adding experiments that explicitly take advantage of the accessibility of the task embedding.
Without such experiments that are more convincing, I do not think the paper meets the bar for acceptance at ICLR.